# Mutation in Sodium-Glucose Cotransporter 2 Results in Down-Regulation of Amyloid Beta (A4) Precursor-Like Protein 1 in Young Age, Which May Lead to Poor Memory Retention in Old Age

**DOI:** 10.3390/ijms21155579

**Published:** 2020-08-04

**Authors:** Keiko Unno, Yoshiichi Takagi, Tomokazu Konishi, Mitsuhiro Suzuki, Akiyuki Miyake, Takumi Kurotaki, Tadashi Hase, Shinichi Meguro, Atsuyoshi Shimada, Sanae Hasegawa-Ishii, Monira Pervin, Kyoko Taguchi, Yoriyuki Nakamura

**Affiliations:** 1Tea Science Center, University of Shizuoka, 52-1 Yada, Suruga-ku, Shizuoka 422-8526, Japan; gp1747@u-shizuoka-ken.ac.jp (M.P.); gp1719@u-shizuoka-ken.ac.jp (K.T.); yori.naka222@u-shizuoka-ken.ac.jp (Y.N.); 2Production Center for Experimental Animals, Japan SLC Incorporated, 85 Ohara, Kita-ku, Hamamatsu, Shizuoka 433-8102, Japan; yoshiichi-takagi@jslc.co.jp; 3Faculty of Bioresource Sciences, Akita Prefectural University, Shimoshinjo Nakano, Akita 010-0195, Japan; konishi@akita-pu.ac.jp; 4Haruno Branch for Experimental Animals, Japan SLC incorporated, 1478 Haruno-cho Ryoke, Tenryu-ku, Hamamatsu, Shizuoka 437-0626, Japan; slcharu-1@jslc.co.jp (M.S.); slcharu-2@jslc.co.jp (A.M.); slcharu@jslc.co.jp (T.K.); 5Research and Development, Kao Corporation, 2-1-3 Bunka, Sumida-ku, Tokyo 131-8501, Japan; hase.tadashi@kao.com; 6Biological Science Research, Kao Corporation, Akabane, Ichikai-machi, Haga-gun, Tochigi 321-3497, Japan; meguro.shinichi@kao.com; 7Faculty of Health Sciences, Kyorin University, 5-4-1 Shimorenjaku, Mitaka, Tokyo 181-8612, Japan; ats7@ks.kyorin-u.ac.jp (A.S.); sanae_ishii@ks.kyorin-u.ac.jp (S.H.-I.)

**Keywords:** senescence-accelerated mouse prone 10, sodium-glucose cotransporter 2, amyloid beta (A4) precursor-like protein 1, memory retention, glucosuria

## Abstract

Senescence-accelerated mouse prone 10 (SAMP10) exhibits cerebral atrophy and depression-like behavior. A line of SAMP10 with spontaneous mutation in the *Slc5a2* gene encoding the sodium-glucose cotransporter (SGLT) 2 was named SAMP10/TaSlc-*Slc5a2^slc^* (SAMP10-ΔSglt2) and was identified as a renal diabetes model. In contrast, a line of SAMP10 with no mutation in SGLT2 (SAMP10/TaIdrSlc, SAMP10(+)) was recently established under a specific pathogen-free condition. Here, we examined the mutation effect in *SGLT2* on brain function and longevity. No differences were found in the survival curve, depression-like behavior, and age-related brain atrophy between SAMP10-ΔSglt2 and SAMP10(+). However, memory retention was lower in SAMP10-ΔSglt2 mice than SAMP10(+). Amyloid beta (A4) precursor-like protein 1 (*Aplp1*) expression was significantly lower in the hippocampus of SAMP10-ΔSGLT2 than in SAMP10(+) at 2 months of age, but was similar at 12 months of age. CaM kinase-like vesicle association (*Camkv*) expression was remarkably lower in SAMP10(+). These genes have been reported to be involved in dendrite function. Amyloid precursor proteins have been reported to involve in maintaining homeostasis of glucose and insulin. These results suggest that mutation in *SGLT2* results in down-regulation of *Aplp1* in young age, which can lead to poor memory retention in old age.

## 1. Introduction

Senescence-accelerated mice (SAMs) were developed by a group at Kyoto University in Japan [1]. Moreover, in 1981 it was reported that inbred senescence-prone (SAMP) strains were developed as models of accelerated senescence and senescence-resistant (SAMR) strains as the normal aging control [2]. Among SAMP strains, SAMP10 has characteristics of brain atrophy, cognitive decline, and depression-like behavior [3,4]. Therefore, SAMP10 mice have been used as a model of neurodegenerative disease similar to SAMP8 [5], which has been widely used as a model for Alzheimer’s disease [6]. In 2005, SAMP10/TaSlc mice maintained under specific pathogen-free (SPF) conditions in Japan SLC (Hamamatsu-city, Shizuoka, Japan) [7] were discovered to excrete glucose in urine. In 2009, a deletion mutation was found in the sodium-glucose cotransporter 2 (*SGLT2*) of SAMP10/TaSlc. Although there were heterozygous mutant mice in the SAMP10/TaSlc line until around 2008, the line has had no heterozygous mice since 2010. The mutation site was identified in 2014 and we previously reported that SAMP10/TaSlc exhibits persistent glucosuria and lowered expression of Slc5a2 [8]. Based on DNA sequencing, we identified a nucleotide deletion in the *Slc5a2* gene of SAMP10/TaSlc. As the *Slc5a2* gene encodes SGLT2, we designated this strain as SAMP10/TaSlc-*Slc5a2^slc^* (SAMP10-ΔSglt2). On the other hand, SAMP10/TaIdr mice, which had been bred at Aichi Prefectural Welfare Development Center since 1998, did not develop glucoseuria and had no mutation in the *Slc5a2* gene. Mutations in the *Slc5a2* gene were shown to occur spontaneously in SAMP10/TaSlc. Thereafter, the line of SAMP10/TaIdr was reestablished under SPF conditions in Japan SLC as SAMP10/TaIdrSlc (SAMP10(+)).

Using SAMP10/TaIdr, which has no mutation in SGLT2, Shimada et al. have reported that neuronal DNA damage [9], loss of synapse [10], impairment of proteasome activity [11], and microglial impairment [12] are involved in age-related neurodegeneration. On the other hand, we have demonstrated additional characteristics in SAMP10/TaSlc (i.e., in SAMP10-ΔSglt2), including increased superoxide generation [13], DNA oxidative damage [14], and a decrease in antioxidative enzymes [15]. We have also reported preventive effects of antioxidative agents such as green tea catechin, β-cryptoxanthin, green soybean extract, and sesamin on neurodegeneration in SAMP10-ΔSglt2 [16,17,18,19,20,21]. Despite these available data, it has not yet been confirmed whether the mutation of SGLT2 has no effect on age-related brain atrophy, lowered learning and memory abilities, and depressive behavior.

In recent years, an increased number of diabetic patients has become a major problem [22]. Based on the specific expression of SGLT2 in the kidney, SGLT2 inhibitors have been demonstrated to be effective for the treatment of patients with type 2 diabetes. Other than the appearance of sugar in urine, there is no particular problem for patients with renal diabetes. However, SGLT2 has recently been shown to express alfa cells in the pancreas [23], as well as in choroid plexus epithelial cells and ependymal cells in the brain [24]. These results suggest that there might be new cautions regarding the use of SGLT2 inhibitors.

In this study, we compared the characteristics between SAMP10-ΔSglt2 and SAMP10(+) and examined the effect of mutation of SGLT2 on cognitive function, brain atrophy, and longevity. As a result, it was found that SAMP10-ΔSglt2 mice had lower memory retention than SAMP10(+) mice. We investigated whether or not Sglt2 mutations affected gene expression in the brain. Using SAMP10-ΔSglt2 mice, studying the relationship between age-related cognitive decline and glucose homeostasis could be a new strategy for understanding diabetes.

## 2. Results

### 2.1. Characteristics of SAMP10-ΔSglt2, SAMP10(+) and SAMR1

Although the median survival time (MST) of SAMR1 was 710.5 days, herein the time was 432 days in SAMP10(+) and 408 day in SAMP10-ΔSglt2. The lifespan of SAMP10 in both lines—SAMP10-ΔSglt2 and SAMP10(+)—was almost the same (*p* = 0.5506) and was significantly (*p* < 0.0001) shorter than that of SAMR1 (Figure 1A). SAMP10-ΔSglt2 body weight was lower than SAMP10(+) up to eight months old. The weight of SAMP10 in both lines was significantly lower than SAMR1 after five months of age, and their weight decreased after 10 months of age (Figure 1B).

Food intake in SAMP10(+) was significantly lower than in SAMP10-ΔSglt2 and SAMR1. By contrast, there was no significant difference in food intake between SAMP10-ΔSglt2 and SAMR1 except at four months of age (Figure 1C). The blood glucose level of SAMP10-ΔSglt2 was significantly lower than in SAMP10(+) and SAMR1 at four and six months of age, but no significant difference was observed after 12 months of age (Figure 1D). The brain weight of three lines postnatally increased up to six months of age and, thereafter, slightly decreased throughout their lifespans. The brain weights in SAMP10-ΔSglt2 and SAMP10(+) were lower than in SAMR1 after six months of age and no significant difference was observed throughout the lifespan between SAMP10-ΔSglt2 and SAMP10(+) (Figure 1E). Age-related brain atrophy was quite similar between SAMP10-ΔSglt2 and SAMP10(+). Urinary glucose was consistently above 500 mg/dL in mice older than 2 months of age when measured with test strip for clinical examination. There was no effect of aging on urinary glucose [8].

### 2.2. Memory Retention and Depression-Like Behavior

Passive avoidance test was used to study the learning and memory of the animals. In the acquisition trial, three lines showed short response latencies and no significant difference was observed within each line. In the retention test conducted 24 h after the acquisition trial, there was no significant difference between SAMP10(+) and SAMR1. On the other hand, SAMP10-ΔSglt2 had significantly shorter retention latencies compared with SAMP10(+) and SAMR1 (Figure 2A), indicating that SAMP10-ΔSglt2 showed lower memory retention than SAMP10(+).

SAMP10-ΔSglt2 and SAMP10(+) showed a marked increase in immobility as compared with SAMR1 at four months of age (Figure 2B). In contrast, no significant difference was found in immobility among each strain at 12 months of age (Figure 2C). Behavioral responses between SAMP10-ΔSglt2 and SAMP10(+) was quite similar at 4 and 12 months of age, confirming both lines exhibited significant behavioral depression even at young age of tail suspension.

### 2.3. Transcriptome and the Levels of Gene Expression

The hippocampus of two-month-old mice of SAMP10-ΔSglt2 and SAMP10(+) was used for analysis. Transcriptome analysis was performed at this age when no morphological changes were observed yet. DNA microarray data were obtained using high-density oligonucleotide microarrays. The top 10 genes that were significantly up- and down-regulated in SAMP10-ΔSglt2 and SAMP10(+) are listed in Table 1. The amyloid beta (A4) precursor-like protein 1 (*Aplp1*) was significantly down-regulated. Aplp1 was essential for proper synapse maintenance [25] and increased neurogenesis [26]. Cysteine rich protein 61 (Cyr61) was needed for dendritic arborization of hippocampal neurons [27] and the expression level was regulated by methylation [28]. On the other hand, CaM kinase-like vesicle-associated (Camkv) was up-regulated in SAMP10-ΔSglt2. The kinase is reported to be required for dendritic spine maintenance [29,30]. Zinc finger protein of the cerebellum 1 (Zic1) is reported to have function in maintaining neural precursor cells in an undifferentiated state [31]. Protein kinase C, delta (Prkcd) has been implicated in regulating hypothalamic glucose homeostasis [32].

### 2.4. Effect of Sglt2 Mutation on Gene Expression in Hippocampus

The expression levels of Aplp1, Cyr61, and Camkv were examined by quantitative real-time reverse transcription PCR (qRT-PCR). The degree of gene expression in the hippocampus of SAMP10-ΔSglt2 was compared with SAMP10(+) and SAMR1, and at 2 and 11–12 months of age to compare whether the changes in the younger ages persist into old age. At two months, the level of Aplp1 was significantly lower in SAMP10-ΔSglt2 than SAMP10(+) and SAMR1. However, the level of SAMP10-ΔSglt2 increased drastically to levels similar to SAMP10(+) at 11–12 months of age (Figure 3). On the other hand, the level of Camkv was significantly lower in SAMP10(+) than SAMP10-ΔSglt2 and SAMR1 at both 2 and 11–12 months. The level of Cyr61 tended to be higher in both SAMP10 than SAMR1 at both 2 and 11–12 months, but there was no difference between SAMP10-ΔSglt2 and SAMP10(+). Individual differences affected the transcriptome data of Cyr61 because the analysis was done using each two samples. Differences in gene methylation may be a cause of individual differences in the expression level of Cyr61 in SAMP10 [28].

Since differences in memory retention were observed in both lines of SAMP10, we compared the expression levels of synaptophysin (Syn) and postsynaptic density 95 (PSD95) as synapse-related proteins. These levels were not significantly different among SAMR1, SAMP10-ΔSGLT2, and SAMP10(+) at both 2 and 11–12 months.

## 3. Discussion

Lines of SAMP10, SAMP10(+) and SAMP10-ΔSglt2 were found to exhibit similar shortened lifespan, age-related brain atrophy, and depression-like behavior. However, there was a difference in memory retention between SAMP10(+) and SAMP10-ΔSglt2. Originally, SAMP10 had age-related decreased memory retention [5], but newly established SPF grade SAMP10 (SAMP10(+)) had a high memory retention ability, similar to SAMR1. The gene background or epigenetic modification of SAMP10(+) may be different from the original SAMP10 (SAMP10/TaIdr). On the other hand, SAMP10-ΔSglt2 showed reduced memory retention ability while aging. The cause of such a difference in memory retention ability was unknown. SAMP10-ΔSglt2 mice had lower body weight and blood glucose levels than SAMP10(+), despite a higher food intake than SAMP10(+). Slight but long-lasting low levels of blood glucose can have some disadvantageous effects on cognitive function. Hypoglycemia has been reported to reduce cognitive function [33,34]. Recently, SGLT2 was reported to be expressed in choroid plexus epithelium epithelial cells and ependymal cells [24], which suggests that glucose uptake from the cerebrospinal fluid to the brain may be reduced. This can be a reason for poor performance during the memory retention test.

Aplp1 and Aplp2 are members of the amyloid precursor protein (APP), which is the source of the neurotoxic amyloid beta (Aβ) peptide involved in Alzheimer’s disease (AD). Although all APP family members have a role in synapse formation and synaptic plasticity, Aplp1 is reported to be especially essential for synapse maintenance [35]. In addition, as a novel function for the APP family, *APP* and *Aplp2* expression has been reported to modulate plasma insulin, glucose concentration, and body weight [25]. Aplp1 may be involved in glucose metabolism as a member of the APP family. Since Aplp1 plays an important role in synapse formation, it is easily predicted that a significant decrease in expression at an early age has an important effect. Despite the increased food intake in SAMP10-ΔSglt2, the blood glucose level was lower in SAMP10-ΔSglt2 than SAMP10(+) at a young age. The altered expression of *Aplp1* with age may be involved in changes in blood glucose levels and body weight. SAMP10-ΔSglt2 is a model of renal diabetes. It is possible to easily put mice in a hypoglycemic state by controlling the food. It also serves as a model for long-term use of SGLT2 inhibitors. In addition, SAMP10-ΔSglt2 may be a useful model for studying the role of Aplp1 in cognition and glucose homeostasis.

We have previously reported that the expression of *Aplp1* was suppressed in aged SAMP10-ΔSglt2 ingested with the green soybean extract. At that time, the decline of cognitive function and Aβ accumulation were suppressed [19]. High expression of *Aplp1* increased Aβ accumulation. However, similar levels of *Aplp1* in both lines of aged SAMP10 (Figure 3) suggested that low level of *Aplp1* at young age was more important for aging-related cognitive decline than Aβ accumulation. It is currently unknown why *Aplp1* expression changes significantly with age. Some abnormality may be occurring in the metabolism or gene expression control of APPs, including *Aplp1*.

Camkv is reported to be an important synaptic protein in maintaining dendritic spines because the knockdown in hippocampal CA1 impairs synaptic transmission and plasticity [29]. Low expression of *Camkv* in SAMP10(+) may be a problem because a precise regulation of Camkv for activity-dependent synthesis and post-translational phosphorylation is critical for dendritic spine maintenance. The level in SAMP10-ΔSGLT2 tended to be higher than SAMR1. The mutation of SGLT2 may cause abnormal regulation of *Camkv*, resulting in high expression and abnormal maintenance of dendrite. The *Camkv* gene is reported to be one of the more promising loci for post-traumatic stress disorder [36]. SAMP10(+) may be a useful PTSD model showing decreased *Camkv* expression.

Camkv phosphorylated by cyclin-dependent kinase 5 causes activation of RhoA, resulting in a loss of dendrite spines [30]. Tight regulation of RhoA activity is crucial for maintaining dendritic spines. The difference between RhoA activity and the expression of *Camkv* and *SGLT2* mutations in SAMP10 strain still need to be investigated. The reason why *Cyr61* increased in both SAMP10 lines is also a potential topic for future study. Pre- and post-synaptic markers, *Syn* and *PSD95*, did not show any difference in mRNA expression between the two lines of SAMP10, but their protein levels need to be investigated.

A detailed research on morphological changes of dendrite in SAMP10 has already been conducted by Shimada et al. [37]. Since SAMP10-ΔSglt2 and SAMP10(+) showed similar brain atrophy (Figure 1E), both lines are expected to show similar morphological changes. However, detailed studies of dendritic morphological changes will be conducted in the near future.

In conclusion, we found that the mutation of *SGLT2* results in down-regulation of *Aplp1* during young age, which can lead to poor memory retention in old age. On the other hand, *Camkv* was up-regulated. In the future, it will be necessary to clarify the significance of SGLT2 expression in the choroid layer in brain and in pancreatic alpha cells, as well as to carefully observe the effect of SGLT2 inhibitors on brain function.

## 4. Materials and Methods

### 4.1. Animals

Male SAMP10/TaSlc (SAMP10-ΔSglt2), SAMP10/TaIdrSlc (SAMP10(+)), and SAMR1/TaSlc (SAMR1) obtained from Japan SLC (Shizuoka, Japan) were bred under SPF conditions in a temperature- and humidity-controlled room with a 12/12-h light/dark schedule (24 ± 1 °C; 45–65% humidity; light period, 08.00–20.00 h). A normal diet (MR-A1; Nosan Corporation, Kanagawa, Japan) and tap water were available ad libitum. Male SAMR1 mice as control mice have normal longevity and a similar genetic background to SAMP10 mice. At the start of the longitudinal study, four-week-old male mice were selected and housed alone per cage, preventing fights. All mice were inspected at least once a day. All study procedures were reviewed and approved by Japan SLC Animal Care and Use Committee and University of Shizuoka Laboratory Animal Care Advisory Committee (approval No. 195241, 9 January 2020). They were in accordance with the guidelines of the US National Institutes of Health for the care and use of laboratory animals.

### 4.2. Measurements of Physiological Parameters, Glucose Levels, and Brain Weight

Mice were weighed, food intake was calculated, and blood glucose levels were measured using a blood glucose meter and test tips (Bayer Yakuhin, Ltd., Osaka, Japan). Measurements of blood glucose level were done from 2 pm to 4 pm at a fixed time. After decapitation, the brain was weighed at 4, 6, 8, 12, and 15 months of age.

### 4.3. Measurements of Behavioral Task

Learning and memory abilities were assessed by acquisition trials and retention tests, respectively, using a passive avoidance system. The passive avoidance response procedure was described in a previous paper [38], wherein a two-compartment step-through passive avoidance apparatus SGS-002 (Muromachi Kikai Co., Ltd., Tokyo, Japan) was used. A 0.5 mA foot shock was applied to the floor grid for 3 s.

Depression-like behavior was assessed as immobility time by the tail suspension test. Each mouse was suspended by the tail for 15 min using a tail suspension apparatus BS-TS2 (Brain Science. Idea. Co., Ltd., Osaka, Japan) and the amount of movement was automatically recorded. Tasks at different ages were done using different groups of mice.

### 4.4. Measurement of DNA Microarray and Principal Component Analyses

Each mouse was used at two months of age. An RNeasy Mini Kit (74104, Qiagen, Valencia, CA, USA) was used for extraction of total RNA from the hippocampus. To synthesize biotinylated cRNA, total RNA was processed using one-cycle target labeling and control reagents (Affymetrix, Santa Clara, CA, USA), and hybridized to a Total RNA Mouse Gene 1.0 ST Array (Affymetrix) with three biological repeats per group. Raw data that were parametrically normalized [39] were statistically tested by two-way ANOVA [40] at *p* < 0.001.

### 4.5. Quantitative Real-Time Reverse Transcription PCR (qRT-PCR)

The hippocampus of mice aged 2 and 11–12 months was used for this analysis. qRT-PCR analysis was performed using the PowerUp™ SYBR™ Green Master Mix (A25742, Applied Biosystems Japan Ltd., Tokyo, Japan) and automated sequence detection systems (StepOne, Applied Biosystems Japan Ltd., Tokyo, Japan). Relative gene expression was measured by previously validated primers for *Aplp1* [41], *Camkv* [29], *Cyr61* [42], *Syn* and *PSD95* [43] genes. cDNA derived from transcripts encoding β-actin was used as the internal control.

### 4.6. Statistical Analyses

Data are expressed as means ± standard error of the mean (SEM). Statistical analyses were performed using GraphPad Prism version 7.0 (GraphPad Software Inc, San Diego, CA, USA). Survival data were analyzed by the log-rank (Mantel–Cox) test and Kaplan–Meier survival curves. The passive avoidance response was compared by one-way analysis of variance (ANOVA) followed by the Kruskal–Wallis test. Other parameters ware analyzed by ANOVA and followed by the Tukey–Kramer method, where *p* < 0.05 was considered statistically significant.

## 5. Conclusions

We found that mutations in *SGLT2* cause down-regulation of *Aplp1* during young age but not old age for SAMP10-ΔSglt2 mice. Since *Aplp1* is essential for synaptic maintenance, the reduced expression may lead to reduced memory retention in old age. On the other hand, *Camkv* was low in SAMP10(+) and slightly higher in SAMP10-ΔSglt2 than SAMR1. Since precise regulation of Camkv is important for maintaining dendritic spines, altered expression of *Camkv* may be associated with depressive behavior. Summarized data is shown in Table 2.

## Figures and Tables

**Figure 1 ijms-21-05579-f001:**
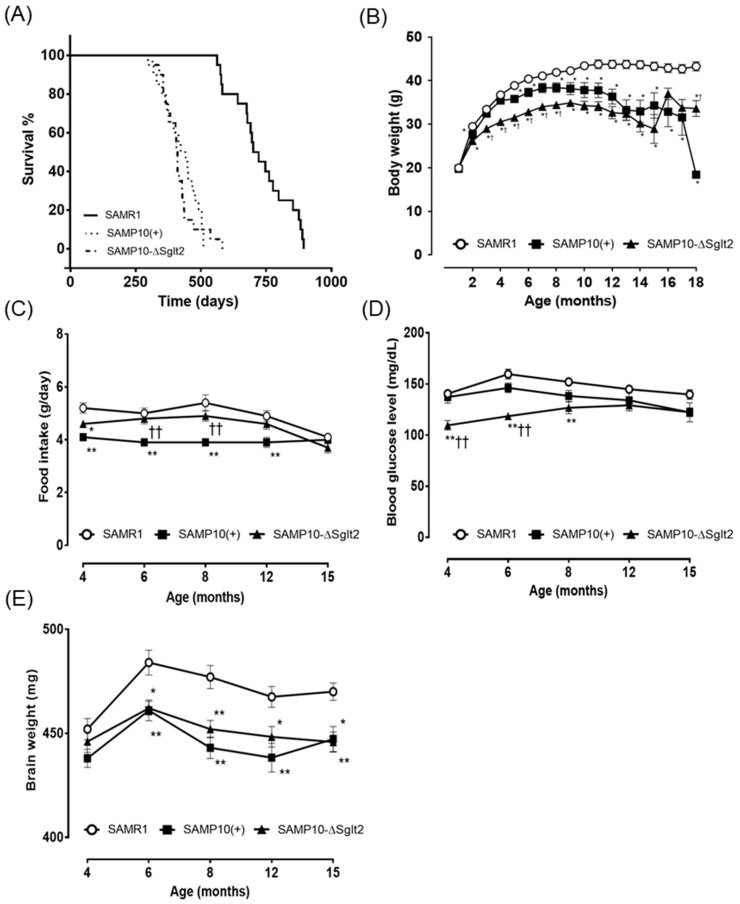
Survival curves (**A**) and body weight (**B**) in male senescence-accelerated mouse prone10 (SAMP10)/TaSlc mice (SAMP10-ΔSglt2), SAMP10/TaIdrSlc (SAMP10(+)) mice, and SAMR1/TaSlc (SAMR1) (*n* = 20 in each group). Data are expressed as mean ± standard error of the mean (SEM) in (**B**). Age-related change in food intake (**C**), blood glucose levels (**D**), and age-related change in brain weight (**E**) in male SAMP10-ΔSglt2, SAMP10(+), and SAMR1 at 4, 6, 8, 12, and 15 months of age. SAMR1 mice: *n* = 10; SAMP10(+) mice: *n* = 7–10; SAMP10-ΔSglt2 mice: *n* = 8–10. Data are expressed as mean ± SEM. * *p* < 0.05 and ** *p* < 0.01 versus SAMR1 mice; ^†^
*p* < 0.05 and ^††^
*p* < 0.01 versus SAMP10(+).

**Figure 2 ijms-21-05579-f002:**
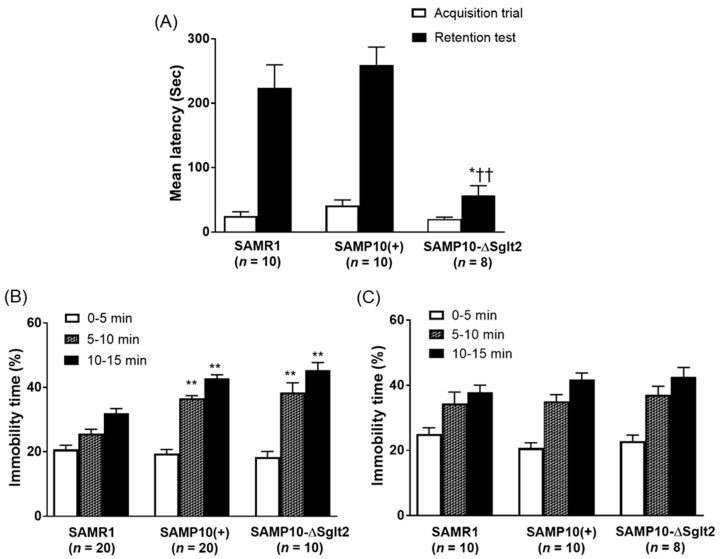
Passive avoidance response at 12 months of age in male SAMP10-ΔSglt2, SAMP10(+) and SAMR1 (**A**). Tail suspension at 4 months of age (**B**) and 12 months of age (**C**) in male SAMP10-ΔSglt2, SAMP10(+) and SAMR1. Data are expressed as mean ± SEM. * *p* < 0.05 and ** *p* < 0.01 versus SAMR1; ^††^
*p* < 0.01 versus SAMP10(+).

**Figure 3 ijms-21-05579-f003:**
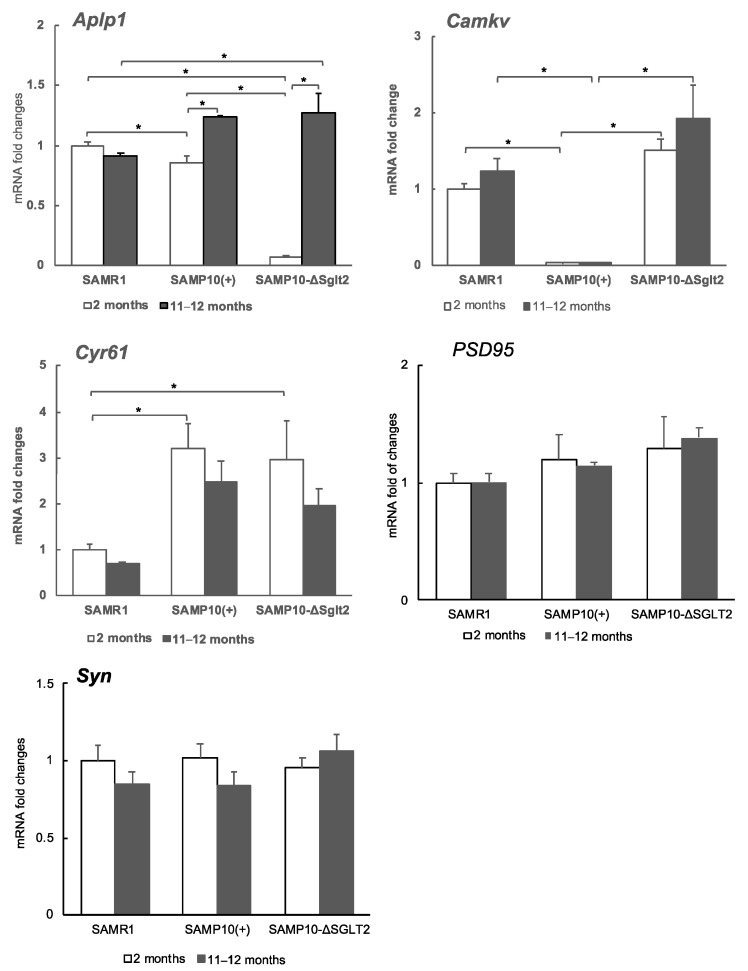
Expression of genes in hippocampi of in male SAMP10-ΔSglt2, SAMP10(+) and SAMR1. The levels of Aplp1, Camkv, Cyr61, PSD95, and Syn were measured at 2 and 11–12 months of age (*n* = 4–6, * *p* < 0.05).

**Table 1 ijms-21-05579-t001:** Down- and up-regulated genes in the hippocampus of SAMP10-ΔSGLT2 compared with SAMP10(+) at 2 months of age.

	Symbol	Full Name	ΔZ	*p*
**Down-Regulated**	Aplp1	amyloid beta (A4) precursor-like protein 1	−1.1688	6.77 × 10^−48^
	Olfr716	olfactory receptor 716	−0.4277	0.0013
	Trav14-1	T cell receptor alpha variable 14-1	−0.5237	0.0031
	Cyr61	cysteine rich protein 61	−0.2115	0.0004
	Ifna12	interferon alpha 12	−0.3784	0.0004
	Sult2a2	sulfotransferase family 2A, dehydroepiandrosterone (DHEA)-preferring, member 2	−0.3743	0.0072
	Pth	parathyroid hormone	−0.2515	0.0014
	LOC100043315	uncharacterized LOC100043315	−0.2768	0.0087
	Rpl28-ps4	ribosomal protein L28, pseudogene 4	−0.2998	0.0024
	Prl2c1	Prolactin family 2, subfamily c, member 1	−0.2691	0.0082
**Up-Regulated**	Camkv	CaM kinase-like vesicle-associated	1.5327	6.73 × 10^−47^
	Mir148b	microRNA 148b	0.4986	0.0003
	Vmn1r177	vomeronasal 1 receptor 177	0.3498	0.0078
	Zic1	zinc finger protein of the cerebellum 1	0.3551	2.67 × 10^−16^
	LOC434035	immunoglobulin kappa-chain VK-1	0.3064	0.0069
	Prkcd	protein kinase C, delta	0.3021	1.93 × 10^−12^
	Aspn	asporin	0.2295	0.0052
	Vmn1r8	vomeronasal 1 receptor 8	0.3163	0.0053
	Tcf7l2	transcription factor 7 like 2, T cell specific, HMG box	0.2341	0.0002
	Calb2	calbindin 2	0.2799	4.73 × 10^−8^
		ΔZ = expression level (SAMP10-∆Sglt2 − SAMP10(+))		

**Table 2 ijms-21-05579-t002:** Characterization of SAMP10-ΔSglt2 and SAMP10(+) compared to SAMR1.

Mouse Line	SAMR1	SAMP10-ΔSglt2	SAMP10(+)
Lifespan	Long	Short	Short
Cerebral atrophy	−	+	+
Depression	−	+	+
Mutation in SGLT2	−	+	−
Glucose in urine	−	+	−
Glucose in blood	Normal	Low in young	Normal
Memory retention	High	Low in aged	High
*Aplp1* in the hippocampus	Normal	Low in young	Normal in young
*Camkv* in the hippocampus	Normal	Slightly high	Low

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
