# Peer review of "Mutation in Sodium-Glucose Cotransporter 2 Results in Down-Regulation of Amyloid Beta (A4) Precursor-Like Protein 1 in Young Age, Which May Lead to Poor Memory Retention in Old Age"

_ijms, 2020, doi:10.3390/ijms21155579_

Round 1

Reviewer 1 Report

Thank you for the opportunity to review this manuscript entitled ‘Mutation in Sodium-Glucose Cotransporter 2 Results in Down-Regulation of Amyloid Beta (A4) Precursor like Protein 1 in Young Age, which may Lead to Poor Memory Retention in Old Age’ by Unno and Colleagues. The manuscript extends on previous findings to examine the mutation effect in SGLT2 in senescence-accelerated mouse prone 10 (SAMP10) on brain function and longevity. The results indicate that mutation in SGLT2 results in down40 regulation of Aplp1 in young age, which can lead to poor memory retention in old age. The manuscript is well written and logical and presents data appropriate for publication in this journal. However, there are some key points that require clarification.

  1. Why is there no age-related glucose in urine in paragraph 2.1? From the introduction it follows that this is an important feature of the SAMP10 / TaSlc mouse strain. It needs to be added.
  2. In clause 2.1. there is no description of age-related blood glucose results. It needs to be added.
  3. Why did you use 2 month old mice for transcriptome analysis (point 2.3)? Are SAMP10 / TaSlc brain changes already recorded at this age? If yes, please indicate this in the text.
  4. How can you explain that for the Cyr61 gene, qRT-PCR analysis (paragraph 2.4.) Did not confirm the results of the transcriptome analysis?
  5. Why were 2-month-old animals used for transcriptome analysis, and 2 and 11-12 months old mice were used for qRT-PCR analysis?
  6. I understand that the hippocampus was used from the qRT-PCR analysis, but this information is not in the text. Please add (paragraph 4.5.).
  7. In the Abstract paragraph, the word “then” is best replaced by “here” (line 33).

Author Response

Comments and Suggestions for Authors

Thank you for the opportunity to review this manuscript entitled ‘Mutation in Sodium-Glucose Cotransporter 2 Results in Down-Regulation of Amyloid Beta (A4) Precursor like Protein 1 in Young Age, which may Lead to Poor Memory Retention in Old Age’ by Unno and Colleagues. The manuscript extends on previous findings to examine the mutation effect in SGLT2 in senescence-accelerated mouse prone 10 (SAMP10) on brain function and longevity. The results indicate that mutation in SGLT2 results in down40 regulation of Aplp1 in young age, which can lead to poor memory retention in old age. The manuscript is well written and logical and presents data appropriate for publication in this journal. However, there are some key points that require clarification.

Why is there no age-related glucose in urine in paragraph 2.1? From the introduction it follows that this is an important feature of the SAMP10 / TaSlc mouse strain. It needs to be added.

Thank you very much for reviewing our manuscript.

We added in 2.1 as follows: Urinary glucose was consistently above 500 mg/dL in mice older than 2 months of age when measured with test strip for clinical examination. There was no effect of aging on urinary glucose (data not shown). (line 150−152)

In clause 2.1. there is no description of age-related blood glucose results. It needs to be added.

We already explained in 2.1about that, but we changed the text a little as follows: The blood glucose level of SAMP10-ΔSglt2 was significantly lower than in SAMP10(+) and SAMR1 at four and six months of age, but no significant difference was observed after 12 months of age (Fig 1D). (line 145−146)

Why did you use 2 month old mice for transcriptome analysis (point 2.3)? Are SAMP10 / TaSlc brain changes already recorded at this age? If yes, please indicate this in the text.

No morphological brain changes in SAMP10/TaSlc have been observed at this age. Since it is considered that the transcriptional changes have already occurred before the morphological changes, transcriptome analysis was performed in this age with no morphological changes. This explanation was added in 2.3. (line 183−184)

How can you explain that for the Cyr61 gene, qRT-PCR analysis (paragraph 2.4.) Did not confirm the results of the transcriptome analysis?

Since there were relatively large individual differences in the expression level of Cyr61, a difference was observed between SAMP10-ΔSglt2 and SAMP10(+) in the transcriptome analysis (n = 2), but was not observed in RT-PCR (n = 6). One reason for the individual deference was explained as follows: Differences in gene methylation may be a cause of individual differences in the expression level of Cyr61 in SAMP10 [28]. (line 208−209)

Therefore, we added in 2.4 that individual differences affected the transcriptome data of Cyr61 because the analysis was done using each two samples (line 206−208).

Why were 2-month-old animals used for transcriptome analysis, and 2 and 11-12 months old mice were used for qRT-PCR analysis?

We wanted to confirm whether the change in the expression level in the younger ages persists into old age. We added this explanation in 2.4 (line 200−201).

I understand that the hippocampus was used from the qRT-PCR analysis, but this information is not in the text. Please add (paragraph 4.5.).

We added that hippocampus was used (line 314).

In the Abstract paragraph, the word “then” is best replaced by “here” (line 33).

Thanks. We changed it.

Reviewer 2 Report

Aim of the authors was to study the specific role of Slc5a2 mutation on age-related cognitive decline in a senescence-prone mouse strain. Although the performed experiments are appropriate and the results obtained are promising, on reading the manuscript something seems to be missing. Therefore I would consider performing further experiments before publication. For example revealing alterations in dendritic spine density or morphology could be the final piece of the puzzle. Although these measurements are not mandatory, they would make the manuscript stronger.

Other points:

In section 2.2. it should be mentioned in one sentence, what each behavioural test was used for. For example: “Passive avoidance test was used to study the learning and memory of the animals.” Furthermore in the Materials and methods section a brief description of both test would be useful. This would clarify for the non-experts, for example how “latency” is related to the memory of the animals.

Title of the vertical axis of Fig2/B has different font than the others.

Title of the 2.3. section is “Transcriptome and the Levels of Gene Expression With Aging”,  however only 2-month-old animals were used in this experiment.

In my opinion, lines 135-142, describing the function of the different genes studied, should be moved to the Discussion section.

It is not clear, why 2-month-old animals were used for the gene-expression studies, while 4- (and 12-) month-old for the behavioral tests.

How the expression level was calculated in the transcriptome analysis? What exactly does it mean that the value of ΔZ is 1.5 for example?

The representation of the RT-PCR results is confusing. According to the Materials and methods, relative gene expression (fold change) is presented in the charts. Which group’s result is the basis of the comparison, against which the relative changes were plotted?

In the discussion:  “In addition, the increased expression of Aplp1 with age may cause a decline in memory ability with age.” This conclusion seems to be uncertain, as the other strain has the same high level of Aplp1 mRNA at the age of 12 months, but with a normal memory retention.

Perhaps the Conclusion section would be in a better place right after the Discussion.

Author Response

Comments and Suggestions for Authors

Aim of the authors was to study the specific role of Slc5a2 mutation on age-related cognitive decline in a senescence-prone mouse strain. Although the performed experiments are appropriate and the results obtained are promising, on reading the manuscript something seems to be missing. Therefore I would consider performing further experiments before publication. For example revealing alterations in dendritic spine density or morphology could be the final piece of the puzzle. Although these measurements are not mandatory, they would make the manuscript stronger.

Thank you so much for reviewing our manuscript. We added about dendritic morphological changes in SAMP10 in the discussion. (line 269−272)

Other points:

In section 2.2. it should be mentioned in one sentence, what each behavioural test was used for. For example: “Passive avoidance test was used to study the learning and memory of the animals.” Furthermore in the Materials and methods section a brief description of both test would be useful. This would clarify for the non-experts, for example how “latency” is related to the memory of the animals.

Thanks for your valuable suggestion. We mentioned the behavioral test in 2.2 and 4.3.

Title of the vertical axis of Fig2/B has different font than the others.

Thanks. We corrected it.

Title of the 2.3. section is “Transcriptome and the Levels of Gene Expression With Aging”,  however only 2-month-old animals were used in this experiment.

Removed “With Aging”.

In my opinion, lines 135-142, describing the function of the different genes studied, should be moved to the Discussion section.

Certainly, there is such an idea. However, if the information for these genes is included here, we thought it would be helpful for readers to understand Table 1.

It is not clear, why 2-month-old animals were used for the gene-expression studies, while 4- (and 12-) month-old for the behavioral tests.

Transcriptome analysis was performed at young age, 2-month-old, since it is thought that transcriptional changes have already occurred before morphological changes.

How the expression level was calculated in the transcriptome analysis? What exactly does it mean that the value of ΔZ is 1.5 for example?

The transcriptome data were normalized according to the character that each data set is distributed log-normally (Ref. 43). Therefore, logarithms of the raw data are z-normalized by using a robust calculation. For example, a raw data x1 is transformed as z1= (log(x1)-log(m))/s, where m is the mean data of the data set and s is the standard deviation, and will be common to all the data sets; in this case, s =0.69.

The difference of levels is estimated by subtraction of the z-scores.

ΔZ = z1– z2 = (log(x1)-log(m1))/s – (log(x2)-log(m2))/s = ( log(x1/m1)/(x2/m2) )/s.

So, this is correlated to log-ratios of raw data. When ΔZ=1.5, log10-ratio is 1.5/0.69 =2.1. This means 150 (=10^2.17) times of expressional difference.

The representation of the RT-PCR results is confusing. According to the Materials and methods, relative gene expression (fold change) is presented in the charts. Which group’s result is the basis of the comparison, against which the relative changes were plotted?

Originally, the standard of comparison was the data of SAMP10-ΔSglt2 at 2 months. However, it was confusing, so we changed the value of SAMR1 at 2 months of age to the standard of comparison.

In the discussion:  “In addition, the increased expression of Aplp1 with age may cause a decline in memory ability with age.” This conclusion seems to be uncertain, as the other strain has the same high level of Aplp1 mRNA at the age of 12 months, but with a normal memory retention.

Thanks for your opinion. The sentence was deleted and another possibility was added as follows: High expression of Aplp1 increased Aβ accumulation. However, similar levels of Aplp1 in both lines of aged SAMP10 suggested that low level of Aplp1 at young age was more important for aging-related cognitive decline than Aβ accumulation. (line 249−251)

Perhaps the Conclusion section would be in a better place right after the Discussion.

We followed the order decided in this journal. Following the comment of other reviewer, the summarized data was added as Table 2 in the conclusion. Therefore, this place may be better.

Reviewer 3 Report

Comments for ijms-850483-peer-review-v1

This study investigated whether the Sglt2 mutations affected gene expression in the brain by using SAMP10-ΔSglt2 mice to study the relationship between age-related cognitive decline and glucose homeostasis. The results showed the blood glucose level of SAMP10-ΔSglt2 was significantly lower than in SAMP10(+) and SAMR1 at a young age. Age-related brain atrophy was quite similar between SAMP10-ΔSglt2 and SAMP10, while the SAMP10-ΔSglt2 showed lower memory retention than SAMP10(+). Behavioral responses between SAMP10-ΔSglt2 and SAMP10(+) were quite similar at 4 and 12 months of age, confirming both lines exhibited significant behavioral depression even at the young age of tail suspension. The authors conclude the mutation of SGLT2 results in the down-regulation of Aplp1 during young age, which can lead to poor memory retention in old age.

Comments

  1. In the introduction, the authors noted that SAMP10 mice have been used as a model of neurodegenerative disease similar to SAMP8, which has been widely used as a model for Alzheimer's disease. However, according to the results of this study, newly established SPF grade SAMP10 (SAMP10(+)) or SAMP10-ΔSglt2 could be used to be what kind of experiment as a suitable animal model? Please explain and clarify the reason.
  2. What the relationships among the phosphorylation of Camkv, cyclin-dependent kinase 5, and RhoA activity? How did they affect brain function?
  3. The authors mentioned that they had found that green soybean extract suppressed the expression of Aplp1 and Aβ accumulation in aged SAMP10-ΔSGLT2 in their previous study. Why did they not evaluate the Aβ accumulation in this study? Did the Aβ accumulation has a difference between these two strain?
  4. The connection among each parameter at young and old age should be explained more clearly.

Author Response

This study investigated whether the Sglt2 mutations affected gene expression in the brain by using SAMP10-ΔSglt2 mice to study the relationship between age-related cognitive decline and glucose homeostasis. The results showed the blood glucose level of SAMP10-ΔSglt2 was significantly lower than in SAMP10(+) and SAMR1 at a young age. Age-related brain atrophy was quite similar between SAMP10-ΔSglt2 and SAMP10, while the SAMP10-ΔSglt2 showed lower memory retention than SAMP10(+). Behavioral responses between SAMP10-ΔSglt2 and SAMP10(+) were quite similar at 4 and 12 months of age, confirming both lines exhibited significant behavioral depression even at the young age of tail suspension. The authors conclude the mutation of SGLT2 results in the down-regulation of Aplp1 during young age, which can lead to poor memory retention in old age.

Comments

In the introduction, the authors noted that SAMP10 mice have been used as a model of neurodegenerative disease similar to SAMP8, which has been widely used as a model for Alzheimer's disease. However, according to the results of this study, newly established SPF grade SAMP10 (SAMP10(+)) or SAMP10-ΔSglt2 could be used to be what kind of experiment as a suitable animal model? Please explain and clarify the reason.

The possibility of SAMP10 as animal models was added to the discussion as follows: SAMP10-ΔSglt2 is a model of renal diabetes. It is possible to easily put mice in a hypoglycemic state by controlling the food. It also serves as a model for long-term use of SGLT2 inhibitors. In addition, SAMP10-ΔSglt2 may be a useful model for studying the role of Aplp1 in cognition and glucose homeostasis. (line 243−246)

SAMP10(+) may be a useful PTSD model showing decreased Camkv expression. (line 261)

What the relationships among the phosphorylation of Camkv, cyclin-dependent kinase 5, and RhoA activity? How did they affect brain function?

The explanation in the discussion was corrected as follows: Camkv phosphorylated by cyclin-dependent kinase 5 causes activation of RhoA, resulting in a loss of dendrite spines [39]. Tight regulation of RhoA activity is crucial for maintaining dendritic spines. (line 262−264)

The authors mentioned that they had found that green soybean extract suppressed the expression of Aplp1 and Aβ accumulation in aged SAMP10-ΔSGLT2 in their previous study. Why did they not evaluate the Aβ accumulation in this study? Did the Aβ accumulation has a difference between these two strain?

Since high Aplp1 expression increased Aβ accumulation, this study did not compare Aβ accumulation between these lines of aged SAMP10 with similar Aplp1 levels. We had thought in the green soybean study that accumulation of Aβ was important for cognitive decline. But this study suggested that proper expression levels of Aplp1 are important, especially at young age. We added this to the discussion. (line 249−251)

The connection among each parameter at young and old age should be explained more clearly.

The connection among them was summarized in Table 2 of the conclusions.

Round 2

Reviewer 3 Report

The authors accomplished with the majority of the recommendations and suggestions, I suggest that the manuscript can be published in ijms.